# Experiences of violence and mental health outcomes among Asian American transgender adults in the United States

**Monideepa B. Becerra**[1,2]*, **Erik J. Rodriquez**[3], **Robert M. Avina**[1,2], **Benjamin J. Becerra**[2,4]*

**1** Department of Health Science and Human Ecology, California State University, San Bernardino, California, United States of America, **2** Center for Health Equity, California State University, San Bernardino, California, United States of America, **3** Division of Intramural Research, National Heart, Lung, Blood Institute, Bethesda, Maryland, United States of America, **4** Department of Information and Decision Sciences, California State University, San Bernardino, California, United States of America

* mbecerra@csusb.edu (MBB); bbecerra@csusb.edu (BJB)

## Abstract

### Purpose

We addressed prevalence and factors associated with mental health outcomes (suicidal behavior and psychological distress) among Asian Americans (AA), who identify as transgender, a key group among sexual and gender minorities that is overlooked and understudied.

### Methods

We used data from 2015 United States Transgender Survey during 2019–2020 with our population as census defined AA. Outcomes included suicidal ideation, suicidal thoughts, and serious psychological distress (SPD). Independent variables included any abuse, partner abuse, bathroom-related abuse, and additional covariates. Adjusted odds ratio and 95% confidence interval (aOR; 95% CI) for each outcome are adjusted for age, marital status, citizenship status, education level, employment status, as well as poverty status.

### Results

Nearly 67% reported experiencing any abuse, 52% reported abuse from romantic/sexual partner(s), while 29% reported harassment/abuse when trying to use bathrooms. Moreover, 82% reported suicidal thoughts, 40% reported suicidal attempts, and 39% had SPD. Results demonstrated that any abuse/violence had higher odds of suicidal thoughts (adjusted odds ratio [aOR] = 2.67, 95% confidence interval (CI):[1.98–3.58], suicidal attempts (aOR = 2.83, 95% CI:[2.18–3.68]), and SPD (aOR = 1.56, 95% CI:[1.20, 2.04]). Abuse from romantic/sexual partners had higher odds of suicidal thoughts (aOR = 2.47, 95% CI:[1.76–3.47]), suicidal attempts (aOR = 2.17, 95% CI:[1.68–2.80]), and SPD (aOR = 2.72, 95% CI:[2.03–3.63]). Experience of harassment/abuse during bathroom use had increased odds of suicidal attempts (aOR = 1.81, 95% CI:[1.41–2.31]).

**Data Availability Statement:** The data underlying the results presented in the study are available from the National Center for Transgender Equality

(NCTE) https://www.ustranssurvey.org/data-requests.

**Funding:** The Divisions of Intramural Research at the National Heart, Lung, and Blood Institute and the National Institutes on Minority Health and Health Disparities, National Institutes of Health provided support for this study to author ER. The funders had no role in study design, data collection and analysis, decision to publish, or preparation of the manuscript.

**Competing interests:** The authors have declared that no competing interests exist.

## Conclusion

Exposure to violence is common among AA transgender individuals and related to negative mental health outcomes. Initiatives to reduce exposure to abuse and providing resources for trauma-informed care are imperative to improve health outcomes.

## Introduction

Nearly one in five U.S. adults live with a mental illness, with varying degree of severity, though lowest prevalence has been noted among Asian adults [1]. Despites such data, the literature notes that Asian Americans have low mental health service utilization despite existing need. For example, using the Asian American Quality of Life survey, Jang et al. [2] noted that while 44% of participants had mental distress, only 23% reported any mental health service utilization. Likewise, in summarizing existing literature, Tung [3] noted that only 8.6% of the Asian American population used any mental health related service, as compared to nearly 18% of the general U.S. population. While such literature provide a mix of self-reported and diagnosis results, the overall trends highlight a mental illness burden in the nation. Furthermore, according to recent reports Asian Americans may often view mental illness as signs of "weakness" or disrespect to religious/spiritual beliefs, and discussion of mental illness is often uncommon due to the need to "save face," thus further stigmatizing mental illness [4,5].

Among the Asian American population, those who identify as a sexual and gender minority (SGM) further share a higher prevalence of mental illness when compared to non-SGM Asian Americans as well as other racial/ethnic SGM populations. For example, according to a recent report, 25% of Asian American SGM adults experienced psychological distress, which was higher than that of any other racial/ethnic SGM group and more than four times higher than Asian Americans who are non-SGM [6]. Further, in a qualitative exploratory study among Filipino SGM population in the U.S., Nadal and Corpus [7] noted that the need to balance multiple identities (cultural and sexual orientation), as well as meeting cultural, religious, and family expectations often cumulatively served as stressors. Such participants also reported experiences of racism from within the SGM population that contributed to such stressors. Likewise, researchers have noted that a cumulative intersection of exogenous and endogenous stressors, such as stigma, culture, internalized stress, as well as poor coping skills and social support, have led to worsening mental and sexual health outcomes of sexual and gender minorities who identify as Asian Americans [8,9]. Additionally, funding analyses have demonstrated that of the more than $100,000 awarded for research on health outcomes for Asian American SGM population, no funding has been allocated for mental health related outcomes for this population, including suicide prevention [6]. Despite such evidence on the higher burden of negative mental health outcomes among Asian American SGM population, little research exists on the factors associated with such outcomes.

An even more vulnerable population among SGM are those who identify as transgender. A major challenge of existing data stems from the aggregation of transgender individuals with that of sexual orientations such as lesbian, gay, and bisexual and referred to as LGBT [10]. This may, in turn, unfairly leave the unique transgender population out of empirical data and/or mask their own set of barriers, regardless of sexual orientation. In fact, population-based surveys among transgender populations remains limited [11,12], with little research on Asian Americans who identify as transgender, though recent reports shed light on such prevalence. For example, results from the National Transgender Discrimination Survey [13] highlighted

that transgender respondents who also identified as Asian American, South Asian, Southeast Asian, or Pacific Islander, had higher suicidal attempt (56%), compared to other transgender respondents (41%) and the general U.S. population (4%).

In general, the literature highlights some of the factors that may put SGM at a higher risk of adverse mental health, which include abuse-related factors such as discrimination, harassment, and violence [12], as well as note a higher rate of suicidal behavior among those who identify as transgender, when compared to other sexual minorities [14]. For example, Su et al. noted that the likelihood of depressive symptoms and suicide attempts were significantly higher among transgender youth as compared to those who did not report being transgender [15]. In a study assessing prevalence of health outcomes among transgender and non-transgender patients in a Massachusetts clinic, Reisner et al. found that transgender patients were more likely to report lifetime suicide ideation and suicide attempts as compared to non-transgender patients, as well as a higher rate of social stressors, including abuse [16]. Likewise, a systematic review addressing the barriers to care noted the fear of stigma, stereotyping, lack of competent health professionals, as well as affordability contributed to transgender and gender non-conforming adults from seeking care [17].

Further, in an assessment of experiences of violence and discrimination among the transgender population, Lombardi and colleagues also noted that half of surveyed participants reported an experience of harassment or violence in their lifetime [18]. Using the Nationwide Inpatient Sample, Hanna et al. reported that compared to encounters with cisgender individuals, the odds of mental illness, including anxiety, depression, and psychosis, were all significantly higher among encounters with transgender individuals [19]. The experiences of abuse and how they relate to adverse mental health outcomes with Asian American transgender population, however, remains limited. As such, in this exploratory study, we aimed to address how various experiences of abuse may impact mental health outcomes among Asian American transgender individuals, with a sub-objective of identifying the type of abuse that had the most severe negative role.

## Methods

### Data source

The 2015 U.S. Transgender Survey (USTS) is a survey conducted by the National Center for Transgender Equality, including a total sample size of 27,715 participants from the United States, DC, and territories. Participants were recruited through purposive and snowball sampling. Survey participants were also informed that "trans" or "transgender" is defined as all trans and non-binary identities for the purposes of the survey and thus was inclusive of identities on the trans spectrum. The survey was also developed with input from experts and advocates. It was administered online to transgender adults in the United States only and was disseminated over a 34-day period through community-based outreach efforts. The survey was made available for web-enabled devices, including, smart phones, tablets, computers, etc. as well as in English and Spanish versions. The survey was made available to all United States' states and territory, as well as military bases, and to those who were at least 18 years of age during the time of the survey. Respondents included those from all 50 states, DC, as well as military bases overseas and American Samoa, Guam, and Puerto Rico. Additional information regarding the study design are available elsewhere [20]. In this study, all sample sizes are weighted, as provided by USTS.

We focused on the Asian American population who were considered transgender in the context of the aforementioned definition used by USTS. USTS provided a six-category recoded variable for self-reported racial/ethnic group based on the American Community Survey

(Alaska Native/American Indian alone, Asian/Native Hawaiian/Pacific Islander, Biracial/Multiracial/Not listed, Black/African American alone, Latino/Latina/Hispanic alone, White/Middle Eastern/North African alone). The Asian American population of this study were defined as those who identified as Asian/Native Hawaiian/Pacific Islander group since only 7.89% were Native Hawaiian/Pacific Islander and thus they were not included in the selection. This study was approved by the Institutional Review Board of California State University, San Bernardino (IRB #IRB-FY2019-192). No competing financial interests exist.

## Measures

Outcome variables for this study included USTS-provided dichotomized variables of suicidal thoughts (also referred to as ideation), suicidal attempt, and serious psychological distress. Suicidal thoughts were coded as ever having serious suicidal thoughts, defined by responding *yes* to either: *At any time in the past 12 months did you seriously think about trying to kill yourself?* or *At any time in your life, have you seriously thought about trying to kill yourself?* Suicidal attempt was coded as ever responding *yes* to: *During the past 12 months, did you try to kill yourself?* or *At any time in your life, did you try to kill yourself?* Finally, serious psychological distress was assessed using the Kessler-6 scale and defined as receiving a score of 13 or higher with a Cronbach's alpha of 0.89 [21].

The independent variables of the study were any abuse/violence, partner abuse/violence, and bathroom use related harassment/abuse. Any abuse/violence was defined as having responded *yes* to any one of the following questions: *In the past year, did anyone verbally harass you for any reason? In the past year, did anyone physically attack you (such as grab you, throw something at you, punch you, use a weapon) for any reason? Have you ever experienced unwanted sexual contact (such as oral, genital, or anal contact or penetration, forced fondling, rape)?* OR *Now just thinking about the past year, have you experienced unwanted sexual contact (such as oral, genital, or anal contact or penetration, forced fondling, rape)?*

Partner abuse/violence was defined as having responded *yes* to any one of the following questions: *Have any of your romantic or sexual partners ever. . .? (a) Tried to keep you from seeing or talking to your family or friends, (b) Kept you from having money for your own use, (c) Kept you from leaving the house when you wanted to go, (d) Hurt someone you love, (e) Threatened to hurt a pet or threatened to take a pet away from you, (f) Wouldn't let you have your hormones, (g) Wouldn't let you have other medications, (h) Threatened to call the police on you, (i) Threatened to "out" you, (j) Told you that you weren't a "real" woman or man, (k) Stalked you, (l) Threatened to use your immigration status against you;* OR *(m) Have any of your romantic or sexual partners ever. . .? (1) Made threats to physically harm you, (2) Slapped you, (3) Pushed or shoved you, (4) Hit you with a fist or something hard, (5) Kicked you, (6) Hurt you by pulling your hair, (7) Slammed you against something, (8) Forced you to engage in sexual activity, (9) Tried to hurt you by choking or suffocating you, (10) Beaten you, (11) Burned you on purpose, (12) Used a knife or gun on you.*

Any harassment/abuse related to bathroom use was coded based on *yes* responses to any of the following questions: *In the past year, did anyone tell or ask you if you were using the wrong bathroom? In the past year, did anyone stop you from entering or deny you access to a bathroom?* OR *In the past year, were you verbally harassed, physically attacked, or experience unwanted sexual contact when accessing or while using a bathroom?*

In addition, the following sociodemographic variables were included in the study as potential covariates: age (18–24 years, 25 to 44 years, 45 years or more), marital status (not currently married, currently married), citizenship status (not a U.S. Citizen, U.S. Citizen), educational attainment (high school or less, some college, associate's degree, bachelor's degree, some

graduate school or more), employment status (not currently employed, currently employed), and poverty level defined at or near poverty (as provided by USTS).

## Data analysis

All descriptive, bivariate, and multivariable analyses utilized survey methods to take into account survey weights and were conducted using SAS v9.4 (SAS Institute, Inc.; Cary, NC). Survey weights were provided to adjust for characteristics of the transgender and U.S. population, including race/ethnicity, age, educational attainment, and income. The survey weights also down-weighted possible respondents who could not be distinguished between 17 and 18 years of age due to their birth year. Descriptive statistics consisted of weighted frequencies (rounded up to nearest whole number, as the weights were provided by USTS as fractions) and percentages. Bivariate survey-weighted chi-square analyses and survey-weighted multivariable logistic regression analyses were used to assess the relationship between abuse, partner abuse, or bathroom problems and outcomes of suicidal ideation, suicidal attempts, and serious psychological distress, with the latter adjusting for the aforementioned demographic variables. Interactions, based on relevant literature, were also assessed between abuse, partner abuse, or bathroom problems with suicidal ideation, suicidal attempts, and serious psychological distress. Sub-analyses were further conducted by specific gender identity and prevalence of abuse types, mental health outcomes, as well as associated regressions. Gender identity provided by USTS included: crossdresser, transgender man, transgender woman, assigned female at birth/ gender queer/gender non-binary (AFAB GQ/NB), assigned male at birth/gender queer/gender non-binary (AMAB GQ/NB). Due to low sample size under in some categories, we categorized the gender identity variable as: transgender man, transgender woman, and other. A p-value less than 0.05 determined a statistically significant difference. In addition, p less than .01 and .001 are also provided in tables to demonstrate strength of statistical significance. All were two-tailed analyses. No multicollinearity was present, indicated by a VIF (Variance Inflation Factor) $<10$.

## Results

Table 1 shows the sociodemographic characteristics of the study population (weighted sample = 1,369) in addition to the prevalence of the key variables of this study. All reported percentages have been weighted. A majority of the population (54%) was between 18–24 years old, were not currently married (89.8%), were U.S. citizens (92.5%), had some college, an associate degree, or a bachelor's degree (77.0%), and were currently employed (66.2%). In addition, 36.9% reported being at or below poverty level.

With respect to abuse-related factors, a majority of our study population reported experiencing any form of abuse/violence (66.9%) and abuse/violence from romantic/sexual partner (52.4%), while approximately 29.5% reported harassment/abuse when trying to use a bathroom. Among mental health outcomes, suicidal thoughts were highly prevalent (81.8%), 40% reported suicidal attempts, and 39.1% had serious psychological distress.

As noted in Table 2, results of survey weighted bivariate analyses demonstrated that the prevalence of suicidal thoughts (87.3% vs. 70.6%), suicidal attempts (48.6% vs. 23.6%), and serious psychological distress (43.5% vs. 30.4%) were significantly higher among those who were exposed to any abuse/violence. Likewise, suicidal thoughts (88.3% vs. 76.5%), suicidal attempts (53.2% vs. 33.3%), and serious psychological distress (45.3% vs. 25.5%) were also higher among participants who reported romantic/sexual partner abuse/violence. Lastly, suicidal attempts (50.8% vs. 35.8%) were significantly higher among those who experienced harassment/abuse while using the bathroom.

**Table 1. Characteristics of study sample (1,369).**

|  | *Weighted sample* | **Weighted percent (%)** |
|---|---|---|
| **Demographic Characteristics** |  |  |
| **Age** |  |  |
| 18 to 24 years | 739 | 54.0 |
| 25 to 44 years | 535 | 39.1 |
| 45 years or more | 96 | 7.0 |
| **Marital status** |  |  |
| Not currently married | 1228 | 89.8 |
| Married | 140 | 10.2 |
| **Citizenship status** |  |  |
| Not a U.S. Citizen | 103 | 7.5 |
| U.S. Citizen | 1266 | 92.5 |
| **Education level** |  |  |
| High school or less | 112 | 8.1 |
| Some college, associate, bachelor's | 1054 | 77.0 |
| Some graduate or more | 204 | 14.9 |
| **Employment status** |  |  |
| Not currently employed | 461 | 33.8 |
| Currently employed | 903 | 66.2 |
| **At or below poverty level** | 483 | 36.9 |
| **Abuse-related factors** |  |  |
| Any abuse/violence | 916 | 66.9 |
| Partner abuse/violence | 585 | 52.4 |
| Harassment/abuse when using bathroom | 404 | 29.5 |
| **Mental Health Outcomes** |  |  |
| Suicidal thoughts | 1120 | 81.8 |
| Suicidal attempt | 550 | 40.3 |
| Serious psychological distress | 528 | 39.1 |

Table 3 displays the results of survey weighted multivariable logistic regression where the independent factors of any abuse/violence, romantic/sexual partner abuse/violence, and harassment/abuse related to bathroom use were assessed against suicidal thoughts, suicidal attempts, and serious psychological distress in nine distinct models. Presence of any abuse/violence was associated with higher suicidal thoughts (adjusted odds ratio [aOR] = 2.67, 95% confidence interval [CI]: [1.98, 3.58]), suicidal attempts (aOR = 2.83, 95% CI: [2.18, 3.68]), as well as SPD (aOR = 1.56, 95% CI: [1.20, 2.04]). Likewise, partner abuse/violence, was associated with increased suicidal thoughts (aOR = 2.47, 95% CI: [1.76, 3.47]), suicidal attempt (aOR = 2.17, 95% CI: [1.68, 2.80]), and SPD (aOR = 2.72, 95% CI: [2.03, 3.63]). In addition, harassment/abuse related to bathroom use also increased the odds of suicidal attempts (aOR = 181, 95%CI: [1.41, 2.31]). Assessment of interactions did not reveal any significant results.

Table 4 demonstrates the results of bivariate analyses of prevalence of each type of abuse and each type of mental health outcomes of interest in the study, by that of specific gender identity. The prevalence of any abuse/violence was significantly higher among those who identified as other (gender queer/gender non-binary/crossdresser). On the other hand, trans men had higher prevalence of partner abuse/violence as well as harassment/abuse related to bathroom use. Likewise, suicidal thoughts and attempts were higher among trans men, while prevalence of serious psychological distress was higher among those who identified as other (gender queer/gender non-binary/crossdresser).

**Table 2. Association between sample characteristics and mental health outcomes of suicidal thoughts, suicidal attempts, and serious psychological distress.**

| | | Suicidal Thoughts (%) | Suicidal Attempt (%) | Serious Psychological Distress (%) |
|---|---|---|---|---|
| **Abuse-related Factors** | | | | |
| Any abuse/violence | | *** | *** | *** |
| | No | 70.6 | 23.6 | 30.4 |
| | Yes | 87.3 | 48.6 | 43.5 |
| Partner abuse/violence | | *** | *** | *** |
| | No | 76.5 | 33.3 | 25.5 |
| | Yes | 88.3 | 53.2 | 45.3 |
| Harassment/abuse related to bathroom use | | | *** | |
| | No | 80.6 | 35.8 | 38.0 |
| | Yes | 84.3 | 50.8 | 41.8 |
| **Demographic Characteristics** | | | | |
| Age | | *** | | *** |
| | 18 to 24 years | 85.32 | 39.93 | 51.0 |
| | 25 to 44 years | 79.46 | 42.57 | 27.1 |
| | 45 years or more | 67.92 | 30.77 | 13.7 |
| Marital status | | ** | | *** |
| | Not currently married | 82.81 | 41.04 | 41.4 |
| | Married | 72.87 | 34.57 | 17.8 |
| Citizenship status | | ** | | |
| | Not a U.S. Citizen | 71.43 | 33.45 | 30.8 |
| | U.S. Citizen | 82.66 | 40.90 | 39.8 |
| Education level | | ** | * | *** |
| | High school or less | 89.86 | 49.17 | 50.8 |
| | Some college, associate, Bachelor's | 82.42 | 40.73 | 42.5 |
| | Some graduate or more | 74.34 | 33.63 | 14.7 |
| Employment status | | * | ** | *** |
| | Not currently employed | 84.87 | 45.58 | 53.3 |
| | Currently employed | 80.15 | 37.52 | 31.6 |
| At or below poverty level | | *** | *** | *** |
| | No | 77.21 | 37.03 | 29.4 |
| | Yes | 89.79 | 49.00 | 57.0 |

Results of survey weighted bivariate analyses (weighted sample = 1,369).

* $p < .05$.

** $p < .01$.

*** $p < .001$.

Table 5 provides the results of regression analysis on the odds of suicidal thoughts, suicidal attempts, and serious psychological distress by presence of each type of mental illness and gender identity. While data on serious psychological distress is limited in its interpretation due to low sample size, the overall trend for suicidal thoughts and attempts are similar for the overall trans population as it was for gender identity-specific sub-analyses. For example, among trans women, men, and those in the other category (non-binary, queer, crossdresser), any abuse/violence experience was significantly associated with suicidal thoughts and attempts. Likewise, a similar significant association was noted between partner abuse and mental health outcomes among each of the gender identity groups.

**Table 3. Adjusted odds ratio (OR) and 95% confidence interval (CI) of mental health outcomes[a], (weighted sample = 1,369).**

|  | Suicidal Thoughts | Suicidal Attempt | Serious Psychological Distress |
|---|---|---|---|
| **Any abuse/violence** |  |  |  |
| Yes vs. No | 2.67 (1.98, 3.58)*** | 2.83 (2.18, 3.68)*** | 1.56 (1.20, 2.04)*** |
| **Partner abuse/violence** |  |  |  |
| Yes vs. No | 2.47 (1.76, 3.47)*** | 2.17 (1.68, 2.80)*** | 2.72 (2.03, 3.63)*** |
| **Harassment/abuse related to bathroom use** |  |  |  |
| Yes vs. No | 1.26 (0.91, 1.74) | 1.81 (1.41, 2.31)*** | 1.15 (0.89, 1.50) |

[a]Models were adjusted for age, marital status, citizenship status, education level, employment status, and poverty level.

*** $p < .001$.

## Discussion

Our study aimed to address a gap in existing research by evaluating the association between exposure to various forms of abuse to that of adverse mental health outcomes, including suicide-related behaviors, among a significantly understudied group: Asian American transgender individuals. The results of our study showed that (a) experiences of abuse, including from a partner, are substantially prevalent among the study participants and (b) these experiences are associated with suicidal thoughts, suicidal attempts, and serious psychological distress.

While literature continues to expand on addressing the prevalence of violence, abuse, and suicidal behaviors among transgender population, there has been little focus on whether abuse-related exposures are related to mental health outcomes, especially among Asian Americans, thus addressing the intersectionality of multiple minority status. In a study based in Northern California among male-to-female transgender women who had a history of sex work, Nemoto et al. reported that more than half of those who were Latina and White reported depression, and factors such as transphobia were significantly related to such an outcome [22]. Likewise, results from the Virginia Transgender Health Initiative Survey from 2005–2006 highlight that suicide attempts were higher among participants who experienced physical and/or sexual violence [23]. Similarly, in a study among transgender population recruited through referral agencies in San Francisco, Clements-Noell et al. noted that among other factors, history of forced sex and gender-based discrimination were independently associated with attempted suicide [24]. Howard et al. [25] further noted in a study in Chicago that among transgender people of color, negative experiences at healthcare settings were driven by discrimination against race/ethnicity and/or gender identity, which further led to participants expecting better treatment if they were cis-gendered or white.

**Table 4. Bivariate analyses of prevalence (%) of each type of abuse and mental health outcomes, by gender identity (weighted sample = 1,369).**

|  | Trans Women | Trans Men | Other |
|---|---|---|---|
| Any abuse/violence*** | 63.1 | 69.7 | **76.4** |
| Partner abuse/violence*** | 59.0 | **62.8** | 57.7 |
| Harassment/abuse related to bathroom use*** | 20.8 | **30.1** | 27.3 |
| Suicidal Thoughts*** | 79.4 | **84.7** | 81.4 |
| Suicidal Attempt*** | 39.9 | **44.9** | 37.2 |
| Serious Psychological Distress*** | 34.4 | 34.8 | **46.8** |

*** p < .001.

**Table 5. Adjusted odds ratio (OR) and 95% confidence interval (CI) of mental health outcomes[a] by gender identity group (weighted sample = 1,369).**

|  | Suicidal Thoughts | Suicidal Attempt | Serious Psychological Distress |
|---|---|---|---|
| **Any abuse/violence** (Yes vs. No) |  |  |  |
| Trans Women | 2.78 (1.64, 4.70)*** | 3.22 (2.01, 5.14)*** | 1.03 (0.64, 1.65) |
| Trans Men | 2.32 (1.20, 4.50)* | 2.31 (1.44, 3.69)*** | 2.47 (1.40, 4.36)**[b] |
| Other | 3.09 (1.96, 4.86)*** | 4.55 (2.76, 7.51)*** | 1.65 (1.09, 2.49)* |
| **Partner abuse/violence** (Yes vs. No) |  |  |  |
| Trans Women | 3.22 (1.77, 5.85)*** | 2.23 (1.37, 3.63)** | 1.61 (0.96, 2.72) |
| Trans Men | 2.98 (1.37, 6.51)** | 3.27 (1.99, 5.40)*** | 5.10 (2.64, 9.86)***[b] |
| Other | 2.13 (1.25, 3.65)** | 2.02 (1.35, 3.03)*** | 3.09 (1.99, 4.79)*** |
| **Harassment/abuse related to bathroom use** (Yes vs. No) |  |  |  |
| Trans Women | 1.25 (0.67, 2.34) | 1.80 (1.09, 2.98)* | 1.28 (0.75, 2.18) |
| Trans Men | 0.92 (0.46, 1.82) | 1.30 (0.81, 2.10) | 1.23 (0.71, 2.12)[b] |
| Other | 1.39 (0.86, 2.24) | 2.27 (1.57, 3.29)*** | 1.13 (0.77, 1.65) |

[a]Models were adjusted for age, marital status, citizenship status, education level, employment status, and poverty level.

[b]Models did not converge due to quasi-complete separation.

*$p<0.05$

**$p<0.01$

*** $p < .001$.

Such observations, coupled with those noted in our study, can inform clinicians and public health practitioners as they design implementation research and plan coordinated care efforts. For example, researchers have called for trauma-informed intersectionality analysis [26] in research among such vulnerable populations. As such, patient engagement during research development and implementation phase may provide the needed community perspective, ensure building of trust and transparency, as well as improve representation. Such analysis provides voice to the often vulnerable and unheard to improve future research and practice. Thus, researchers addressing mental health outcomes of SGM population could benefit from evaluating the cultural norms, beliefs, associated- stigma among Asian Americans related to both mental health and SGM status and how they additive impact on health outcomes. Likewise, such populations with history of trauma, such as experiences of violence, can further have heightening worsening outcomes and research that takes into account the cumulative and potentially dose-response based impact on outcomes are needed.

Furthermore, while national studies comparable to our assessment among Asian Americans are lacking, the high prevalence of abuse and suicide-related behaviors in our study population, in addition to national data noting higher rates of suicidal attempt among Asian American transgender populations, compared to their non-Asian American transgender counterparts [13], bring to attention the imperative need for violence prevention and health promotion efforts to alleviate such a burden that take into account cultural barriers in the population.

Examples of putative interventions based on the association noted in our study include cultural competency training and coordinated care that encourage cross-collaboration between public health, allied health, and healthcare systems. Here in lies the opportunity provided by the Patient Protection and Affordable Care Act of 2010, that calls for a Community Health Needs Assessment every three years by non-profit hospitals and for key interventions to be implemented based on the results of such an assessment [27]. A key opportunity here is to integrate professionals for multiple fields as well as community leaders from the Asian American and transgender communities to serve as bridges to assess the prevalence of transgender

identity and needs of the transgender community, such as those related to abuse and mental health, in the service areas of such hospitals as well as to ensure targeted programs are implemented that take into account the unique barriers faced by this vulnerable group.

In addition, improving training and diversity of primary care providers can further address the barriers such populations face. For example, addressing exposure to abuse during primary care screening could further serve as the first step in ensuring suicide prevention in such a vulnerable population and our study results can inform the design of future implementation work that contributes to evidence-based practice. In recent years, many health and related professionals have enhanced their curriculum to integrate course content related to SGM health disparities [28,29]. As such, assessment of such best practices on curriculum updates and how they impact care of SGM populations may provide insight into further evidence-based practice. Further, as noted by participants in a study by Howard et al. [25], having providers of similar racial/ethnic background as well as those who respect patient's gender identity can further serve as a critical need to improve mental health outcomes of such a vulnerable population.

Finally, the Healthy People initiative calls for the improvement of health outcomes of SGM. Yet, with the aggregation of transgender population in the often-used terminology of LGBT, most of the current empirical evidence has not disaggregated this unique population and the barriers they face. This is further coupled with the fact that very little evidence exists for Asian American transgender population, which is often attributed to cultural barriers related to discussion of mental health and sexual orientation [30,31]. As such, a national paradigm shift in research to consider transgender (T) health independent of sexual orientation (LGB) are critical to highlight the importance of addressing the barriers that transgender individuals face.

A sub-analysis of our study population also highlighted that specific gender identities within the trans spectrum have differing prevalence of abuse experiences and mental health outcomes. For example, while prevalence was high among various subgroups, trans men reported a higher prevalence of suicidal thoughts and attempts, when compared to trans women and other. Likewise, trans men had higher bathroom use related harassment as compared to the other groups. We hypothesize that potential resources for trans men support groups may be of lower availability and thus could be contributing to such outcomes. Likewise, gender queer/non-binary may not find easily accessible resources to their unique needs, including social acceptance of they/them/their pronoun, and thus have higher serious psychological distress. A current study reported differences in anxiety and depression among such groups, though results are limited to college students only [32]. These hypotheses need further testing to explore in depth the differing experiences of trans men, when compared to other gender identity, which has been preliminarily highlighted in the current literature [33].

## Limitations

The results of this study should be interpreted in the context of its limitations. Cross sectional data is subject to the lack of causal or temporal relationship and longitudinal results are needed to implement life course health promotion efforts to improve mental health outcomes of the population. Likewise, such survey data is prone to recall bias and given the sensitive questions related to abuse, social desirability bias is inherent. In addition, not all potential confounders were available in this study as we were limited by availability in the data set. The potential role of acculturative stress on mental health among Asian American transgender individuals would be valuable to explore in future studies. The USTS data collection also posits some limitation to the overall study. For example, researchers have reported that some results found in surveys employing nonprobability sampling method, such as USTS, have not been replicated in their counterparts using probability sampling method, such as BRFSS [34]. However, this could be

putatively attributed to social desirability bias where SGM populations maybe reluctant to disclose their status, as seen in a study in Canada [35], as well as the substantial low representation of non cis-gendered adults in such probability sample-based survey due to its reliance on random selection. Given the low prevalence of those who identify as transgender or gender nonconforming, probability sampling methods may not be ideal, without oversampling strategies, to ensure adequate representation.

## Conclusion

Notwithstanding such limitations, the results of our study highlight the role of experiences of abuse and its relation to adverse mental health outcomes among one of the most vulnerable and overlooked groups: Asian American transgender individuals. Our results highlight the importance of such experiences among Asian American transgender individuals and can be used to inform how healthcare and related professionals, such as mental healthcare professionals as well as public health educators, assess the most at-risk patients. Furthermore, such results provide the foundation for healthcare and equity researchers to address the growing need for research in the area of addressing the stigmatized mental illness and needs among the population, especially those who are of SGM status.

## Acknowledgments

The authors would like to acknowledge U.S. Transgender Survey for providing the data.

## Author Contributions

**Conceptualization:** Monideepa B. Becerra.

**Data curation:** Monideepa B. Becerra.

**Formal analysis:** Benjamin J. Becerra.

**Investigation:** Monideepa B. Becerra.

**Methodology:** Monideepa B. Becerra, Benjamin J. Becerra.

**Project administration:** Monideepa B. Becerra.

**Supervision:** Monideepa B. Becerra.

**Validation:** Erik J. Rodriquez.

**Writing – original draft:** Monideepa B. Becerra, Erik J. Rodriquez, Robert M. Avina, Benjamin J. Becerra.

**Writing – review & editing:** Monideepa B. Becerra, Erik J. Rodriquez, Robert M. Avina, Benjamin J. Becerra.

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
