## [Decision Letter · Decision Letter 0]

6 Nov 2020

PONE-D-20-21733

Experiences of violence and mental health outcomes among Asian American transgender adults in the United States

PLOS ONE

Dear Dr. Becerra,

Thank you for submitting your manuscript to PLOS ONE. After careful consideration, we feel that it has merit but does not fully meet PLOS ONE’s publication criteria as it currently stands. Therefore, we invite you to submit a revised version of the manuscript that addresses the points raised during the review process.

There are a number of methodological issues to be addressed, including:

1. Please specify the number of participants included in your sample for analysis - this currently only to be provided as 7.89%

2. Please add more information on how the survey was delivered (online? face to face?), how were participants recruited? What was the eligibility criteria (who was targeted)? 

3. I have some issues with how the outcomes were defined; the  authors state that the outcome variables include dichotomized versions of suicidal ideation, suicidal thoughts, and serious psychological distress - however, only suicide thoughts are reported on (not ideation) - how do thoughts and ideation differ in this study (they usually mean the same thing)? This requires further clarification. Moreover, what instruments were used to ask participants about suicidal ideation and attempts? Were they scales, were they validated? If they were scales, I would query why the authors recoded them into dichotomous variables when a continuous score would have been clinically more significant. If authors did recode scales into dichotomous variables, how was the decision made as to what constitutes a 'yes' answer and what is a 'no' answer? Much more detail is needed. 

4. I similarly question the authors decision to create over simplistic 'yes/no' variables to assess experience of violence, when there appears to be rich data on different types of violence; there is literature to support that different types of violence have different impacts on mental health, and it would be a genuine value add of this study to examine the experience of different types of violence on suicide outcomes in this transgender population. I would strongly suggest some re analysis to provide a more nuanced understanding of the effects of types of violence on suicidal thoughts and attempts. 

We look forward to receiving your revised manuscript.

Kind regards,

Michelle Tye, Ph.D.

Academic Editor

PLOS ONE

Journal Requirements:

"This work was partially supported by the Divisions of Intramural Research at the National Heart, Lung, and Blood Institute and the National Institutes on Minority Health and Health Disparities, National Institutes of Health."

Reviewers' comments:

Reviewer's Responses to Questions

**Comments to the Author**

1. Is the manuscript technically sound, and do the data support the conclusions?

Reviewer #1: Yes

Reviewer #2: Yes

2. Has the statistical analysis been performed appropriately and rigorously? 

Reviewer #1: Yes

Reviewer #2: Yes

3. Have the authors made all data underlying the findings in their manuscript fully available?

Reviewer #1: No

Reviewer #2: Yes

4. Is the manuscript presented in an intelligible fashion and written in standard English?

Reviewer #1: Yes

Reviewer #2: Yes

5. Review Comments to the Author

Reviewer #1: Experiences of violence and mental health outcomes among Asian American transgender adults in the United States

I agree that Asian American transgender individuals are an understudied population, in general, and in the context of negative mental health outcomes including suicidal ideation, suicidal thoughts, and serious psychological distress (SPD). This is novel and important work. This was a secondary data analysis of restricted-use national data.

Abstract

• Methods: can you identify how you subsetted to Asian Americans? How was this assessed in the 2015 USTS? Who is included in this ethnic group?

• Methods/Results: Improve notation for aOR and 95% CI for consistency. For example, define the format first/once in the methods then use it consistently in the results. For example:

o Methods: Adjusted odds ratio and 95% confidence interval (aOR; 95% CI) for each outcome will be adjusted for age, marital status, citizenship status, education level, employment status, and poverty level.

o Results: Experience of harassment/abuse during bathroom use had increased odds of suicidal attempts (1.81; 1.41, 2.31).

Manuscript

• Line 59-60: Does the literature refer to diagnosed cases or self-report?

• Can you speak to stigma for seeking mental health services and care?

• Line 70-71: same question as above. Does the literature refer to diagnosed cases or self-report?

• Line 96: Is the paragraph starting on this line refer to Asian Americans or transgender people nationally

• Line 155 and 164: Why is “OR” in caps?

• Line 185: Improve the statement for alpha or significance level. It is not clear what you are denoting. Perhaps you can say that “the significance level of .05 was used” or “A p-value less than 0.05 will determine a statistically significant difference”. You use other significance levels in the tables. You should mention these here as well.

• Methods/Results: as indicated for the abstract, improve notation for aOR and 95% CI for consistency. For example, define the format first/once in the methods then use it consistently in the results.

• Results: you are referring to “significantly higher “rates. Did you conduct one-tail or two tailed statistical tests? Your method section does not indicate the bivariate tests used.

• Results: Be consistent with rounding for percentages reported though this section including tables.

• Results: In tables, improve title and include sample sizes in the title or somewhere in the table. Distinguish headers from variables. Perhaps consider moving the asterisks denoting significant findings.

• Table 2: you have a symbol in front of “~U.S. Citizen” that you did not define at the footer of the table.

• Conclusion: this section lacks detail particularly to what the results indicate. what type of clinicians are your referring to? What type of assessments are your referring to?

• IRB approval: IRB review is mentioned in the acknowledgement section but not in the methods.

Reviewer #2: The manuscript, “Experiences of violence and mental health outcomes among Asian American transgender adults in the United States,” describes a study examining experiences of Asian American transgender adults in terms of violence and mental health using the 2015 US Transgender Survey. Key findings included high rates of minority stress (discrimination and abuse) as well as negative mental health outcomes, such as suicide ideation and distress. Authors highlight the need for reducing exposure to negative outcomes and greater use of trauma informed care approaches in healthcare settings. Several strengths of this study include novel topic and potential for future research. Despite several strengths, there are a number of research and conceptual issues that if addressed, would significantly improve the manuscript. These are explained below in no particular order.

1. It would be beneficial to readers to learn more about potential research on mental health of Asian American transgender and additional gender minority populations. For instance, several articles should be considered:

Ching TH, Lee SY, Chen J, So RP, Williams MT. A model of intersectional stress and trauma in Asian American sexual and gender minorities. Psychology of violence. 2018 Nov;8(6):657.

Operario D, Nemoto T. Sexual risk behavior and substance use among a sample of Asian Pacific Islander transgendered women. AIDS Education & Prevention. 2005 Oct 1;17(5):430-43.

Snow A, Cerel J, Loeffler DN, Flaherty C. Barriers to mental health care for transgender and gender-nonconforming adults: a systematic literature review. Health & social work. 2019 Aug 2;44(3):149-55.

Howard SD, Lee KL, Nathan AG, Wenger HC, Chin MH, Cook SC. Healthcare experiences of transgender people of color. Journal of general internal medicine. 2019 Oct 1;34(10):2068-74.

2. Methods: Consider providing Cronbach alpha for the Kessler-6 scale for study sample.

3. Table 2 needs a more descriptive/detailed title.

4. It appears that chi-square tests were run to test potential associations. If this represents potential prevalence by factors that are weighted, authors might consider putting the 95% Confidence Intervals

5. For logistic regression, authors may want to present findings in terms of prevalence ratios.

6. Did authors consider testing any interactions with demographics (age, education level, employment, poverty) as it appears these could represent intersectionality. Also may want to consider differences by gender identity (e.g., non-binary vs. gender male/female identifying groups).

7. It appears that additional limitations of the 2015 USTS survey were not mentioned. Authors should consider additional concerns in terms of non-probability sample. See the following for additional limitations:

Henderson, E.R., Blosnich, J.R., Herman, J.L. and Meyer, I.H., 2019. Considerations on sampling in transgender health disparities research. LGBT health, 6(6), pp.267-270.

8. The discussion needs further development, such as other studies in cisgender sexual minorities and cisgender Asian Americans with regards to experiencing abuse and risk of suicide/mental health distress.

9. Discussion should touch how healthcare needs and tailoring of trauma-informed care that considers intersectionality, such as identifying as both Asian American and a gender minority.

10. It might be helpful to readers to further tease of unique cultural concerns for Asian American gender minorities as well as details on how research can be more inclusive of these communities. It might be helpful to expand on any findings in terms of care and needed research on immigration/acculturation and gender minorities.

6. PLOS authors have the option to publish the peer review history of their article (what does this mean?). If published, this will include your full peer review and any attached files.

Reviewer #1: No

Reviewer #2: **Yes: **Jason Flatt

---

## [Author Response · Author response to Decision Letter 0]

15 Dec 2020

Response to reviews:

Thank you for the comprehensive feedback that allows our manuscript to be improved. We have addressed each of the comments below, added content in the manuscript/tables, and have marked them as track changes as well. We have bolded the responses below for ease of finding. 

Review 1

1. It would be beneficial to readers to learn more about potential research on mental health of Asian American transgender and additional gender minority populations. For instance, several articles should be considered:

Ching TH, Lee SY, Chen J, So RP, Williams MT. A model of intersectional stress and trauma in Asian American sexual and gender minorities. Psychology of violence. 2018 Nov;8(6):657.

Operario D, Nemoto T. Sexual risk behavior and substance use among a sample of Asian Pacific Islander transgendered women. AIDS Education & Prevention. 2005 Oct 1;17(5):430-43.

Snow A, Cerel J, Loeffler DN, Flaherty C. Barriers to mental health care for transgender and gender-nonconforming adults: a systematic literature review. Health & social work. 2019 Aug 2;44(3):149-55.

Howard SD, Lee KL, Nathan AG, Wenger HC, Chin MH, Cook SC. Healthcare experiences of transgender people of color. Journal of general internal medicine. 2019 Oct 1;34(10):2068-74.

Response: Our second paragraph addressed the current literature (peer-reviewed and others) on mental health burden Asian American sexual and gender minorities. We have further expanded it by including additional references from those noted above. They have been included in the introduction section (Ching et al., Operaio and Nemoto, Snow et al.) as well as in the discussion section (Howard et al.), in addition to a few others. 

2. Methods: Consider providing Cronbach alpha for the Kessler-6 scale for study sample. 

Response: The Cronbach alpha of .89 has been added in the methods section.

3. Table 2 needs a more descriptive/detailed title.

Response: We have added a more descriptive title that provides details on the context (including analysis) of the table results. 

4. It appears that chi-square tests were run to test potential associations. If this represents potential prevalence by factors that are weighted, authors might consider putting the 95% Confidence Intervals 

Response: The weights applied in this study were used to reduce bias for demographic representation. Since this was a non-probability sample, confidence intervals may not be an appropriate measure of reliability since their estimates are inherently biased from non-random sampling. We have stated the limitations of a non-probability sample in our discussion.

5. For logistic regression, authors may want to present findings in terms of prevalence ratios. 

Response: We did consider reporting prevalence ratio, but during the literature review, we noted that majority of the studies used odds ratio. In order to be utilized for comparison purposes, as well as putative meta analysis by other researchers, we wanted to remain consistent. The interpretation, however, does provide better context on the data. 

6. Did authors consider testing any interactions with demographics (age, education level, employment, poverty) as it appears these could represent intersectionality. Also may want to consider differences by gender identity (e.g., non-binary vs. gender male/female identifying groups).

Response: We checked interactions for based on relevant literature and it is noted in the methods. Survey participants were informed that “trans” or “transgender” is defined as all trans and non-binary identities for the purposes of the survey. As such, the data is inclusive of the different identities. However, per the recommendation, we have added a sub-analysis in our study as part of Tables 4 and 5. 

7. It appears that additional limitations of the 2015 USTS survey were not mentioned. Authors should consider additional concerns in terms of nonprobability sample. See the following for additional limitations: 

Henderson, E.R., Blosnich, J.R., Herman, J.L. and Meyer, I.H., 2019. Considerations on sampling in transgender health disparities research. LGBT health, 6(6), pp.267-270. 

Response: We have included content from this study as well as additional to address the limitations. 

8. The discussion needs further development, such as other studies in cisgender sexual minorities and cisgender Asian Americans with regards to experiencing abuse and risk of suicide/mental health distress. 

Response: We included cisgender focused studies in introduction and discussion, but limited to ensure the focus remains on the need for such among the SGM population, especially Asian Americans transgender populations.

9. Discussion should touch how healthcare needs and tailoring of trauma-informed care that considers intersectionality, such as identifying as both Asian American and a gender minority. 

Response: We have added context related to such analysis in the discussion along with the following comment’s content as well.

10. It might be helpful to readers to further tease of unique cultural concerns for Asian American gender minorities as well as details on how research can be more inclusive of these communities. It might be helpful to expand on any findings in terms of care and needed research on immigration/acculturation and gender minorities.

Response: We have added context related to addressing the unique cultural factors in the discussion.

Review 2

1. Please specify the number of participants included in your sample for analysis - this currently only to be provided as 7.89%

Response: We have added the sample in the results section.

2. Please add more information on how the survey was delivered (online? face to face?), how were participants recruited? What was the eligibility criteria (who was targeted)? 

Response: This has been added in the data source section of the methods.

3. I have some issues with how the outcomes were defined; the authors state that the outcome variables include dichotomized versions of suicidal ideation, suicidal thoughts, and serious psychological distress - however, only suicide thoughts are reported on (not ideation) - how do thoughts and ideation differ in this study (they usually mean the same thing)? This requires further clarification. 

Moreover, what instruments were used to ask participants about suicidal ideation and attempts? Were they scales, were they validated? If they were scales, I would query why the authors recoded them into dichotomous variables when a continuous score would have been clinically more significant. If authors did recode scales into dichotomous variables, how was the decision made as to what constitutes a 'yes' answer and what is a 'no' answer? Much more detail is needed. 

Response: We agree that more detailed content would be valuable. However, the USTS data provides the dichotomized variables as utilized in this study and others. However, the information on instruments for suicidal thoughts, suicidal attempts, and SPD, are added in the measures section. 

4. I similarly question the authors decision to create over simplistic 'yes/no' variables to assess experience of violence, when there appears to be rich data on different types of violence; there is literature to support that different types of violence have different impacts on mental health, and it would be a genuine value add of this study to examine the experience of different types of violence on suicide outcomes in this transgender population. I would strongly suggest some re analysis to provide a more nuanced understanding of the effects of types of violence on suicidal thoughts and attempts. 

Response: USTS has four types of abuses included: past year verbal harassment, past year physical attack, past year unwanted sexual contact, and ever experience of unwanted sexual contact. While we acknowledge that verbal versus physical versus sexual can have differing outcomes, our goal was to explore any experiences of violence and thus we categorized them as any abuse. With a larger sample size, further disaggregation may be of value. However, we did add additional types of abuse, including partner and harassment during bathroom use.

---

## [Decision Letter · Decision Letter 1]

11 Jan 2021

PONE-D-20-21733R1

Experiences of violence and mental health outcomes among Asian American transgender adults in the United States

PLOS ONE

Dear Dr. Becerra,

Thank you for submitting your manuscript to PLOS ONE. After careful consideration, we feel that it has merit but does not fully meet PLOS ONE’s publication criteria as it currently stands. Therefore, we invite you to submit a revised version of the manuscript that addresses the points raised during the review process.

The authors have adequately addressed the first set of revisions, there are some additional queries below that need to be addressed, and which should significantly strengthen the manuscript. 

We look forward to receiving your revised manuscript.

Kind regards,

Michelle Tye, Ph.D.

Academic Editor

PLOS ONE

Reviewers' comments:

Reviewer's Responses to Questions

**Comments to the Author**

1. If the authors have adequately addressed your comments raised in a previous round of review and you feel that this manuscript is now acceptable for publication, you may indicate that here to bypass the “Comments to the Author” section, enter your conflict of interest statement in the “Confidential to Editor” section, and submit your "Accept" recommendation.

Reviewer #1: (No Response)

2. Is the manuscript technically sound, and do the data support the conclusions?

Reviewer #1: Yes

3. Has the statistical analysis been performed appropriately and rigorously? 

Reviewer #1: I Don't Know

4. Have the authors made all data underlying the findings in their manuscript fully available?

Reviewer #1: Yes

5. Is the manuscript presented in an intelligible fashion and written in standard English?

Reviewer #1: Yes

6. Review Comments to the Author

Reviewer #1: I agree that Asian American transgender individuals are an understudied population, in general, and in the context of negative mental health outcomes including suicidal ideation, suicidal thoughts, and serious psychological distress (SPD). This is novel and important work. This was a secondary data analysis of restricted-use national data.

Abstract

• Methods: Can you identify how you subsetted to Asian Americans? How was this assessed in the 2015 USTS? Who is included in this ethnic group?

• Methods/Results: Improve notation for aOR and 95% CI for consistency. For example, define the format first/once in the methods then use it consistently in the results. For example:

o Methods: Adjusted odds ratio and 95% confidence interval (aOR; 95% CI) for each outcome will be adjusted for age, marital status, citizenship status, education level, employment status, and poverty level.

o Results: Experience of harassment/abuse during bathroom use had increased odds of suicidal attempts (aOR: 1.81; 95% CI: 1.41, 2.31).

Manuscript

• Line 59-60: Does the literature refer to diagnosed cases or self-report?

• Can you speak to stigma for seeking mental health services and care?

• Line 70-72: same question as above. Does the literature refer to diagnosed cases or self-report?

• Line 155 and 157: Why is “OR” in caps? Perhaps consider listing the measures without stating the entire question asked, separated with semicolons. For example, instead of “In the past year, did anyone verbally harass you for any reason?”, you can indicate “verbal harassment for any reason in past year”.

• Line 211: Improve the statement for alpha or significance level. It is not clear what you are denoting. Perhaps you can say that “The significance level of .05 was used” or “A p-value less than 0.05 determined a statistically significant difference”. You use other significance levels in the tables. You should mention these here as well.

• Methods/Results: as indicated for the abstract, improve notation for aOR and 95% CI for consistency. For example, define the format first/once in the methods then use it consistently in the results.

• Results: you are referring to “significantly higher “rates. Did you conduct one-tail or two tailed statistical tests? Your method section does not indicate the bivariate tests used (Line 200).

• Results: Be consistent with rounding for percentages reported though this section including tables.

• Results: In tables, improve title and include sample sizes in the title or somewhere in the table. Distinguish headers from variables. Perhaps consider moving the asterisks denoting significant findings.

• Tables: Consider justifying to the right for the numeric columns.

• Table 2: You have a symbol in front of “~U.S. Citizen” that you did not define at the footer of the table.

• Conclusion: this section lacks detail particularly to what the results indicate. What type of clinicians are your referring to? What type of assessments are your referring to?

7. PLOS authors have the option to publish the peer review history of their article (what does this mean?). If published, this will include your full peer review and any attached files.

Reviewer #1: No

---

## [Author Response · Author response to Decision Letter 1]

10 Feb 2021

Response to reviews:

Thank you for the reviews of our manuscript. We have addressed all the reviews and responses are noted below in bold, for ease of finding.

Abstract

• Methods: Can you identify how you subsetted to Asian Americans? How was this assessed in the 2015 USTS? Who is included in this ethnic group?

Response: We included this in the methods section of the manuscript, instead of abstract, to ensure we remain within the word limit. However, we added a statement to clarify it was census defined as well. 

• Methods/Results: Improve notation for aOR and 95% CI for consistency. For example, define the format first/once in the methods then use it consistently in the results. For example:

o Methods: Adjusted odds ratio and 95% confidence interval (aOR; 95% CI) for each outcome will be adjusted for age, marital status, citizenship status, education level, employment status, and poverty level.

o Results: Experience of harassment/abuse during bathroom use had increased odds of suicidal attempts (aOR: 1.81; 95% CI: 1.41, 2.31).

Response: We have updated the methods section in the abstract accordingly.

We have updated the results section as well to be consistent and note aOR in each case. 

Manuscript

• Line 59-60: Does the literature refer to diagnosed cases or self-report?

• Can you speak to stigma for seeking mental health services and care?

• Line 70-72: same question as above. Does the literature refer to diagnosed cases or self-report?

Response: This has been updated in the instruction section. We also added the importance of addressing stigma in the conclusion. 

• Line 155 and 157: Why is “OR” in caps? Perhaps consider listing the measures without stating the entire question asked, separated with semicolons. For example, instead of “In the past year, did anyone verbally harass you for any reason?”, you can indicate “verbal harassment for any reason in past year”.

Response: In the previous revision cycle, we were asked to list the full question. However, we have updated the structure to read with more ease and not capitalized the OR. 

• Line 211: Improve the statement for alpha or significance level. It is not clear what you are denoting. Perhaps you can say that “The significance level of .05 was used” or “A p-value less than 0.05 determined a statistically significant difference”. You use other significance levels in the tables. You should mention these here as well.

Response: We’ve updated this section to read more clearly as well as add the .01 and .001 mentioned in the tables. 

• Methods/Results: as indicated for the abstract, improve notation for aOR and 95% CI for consistency. For example, define the format first/once in the methods then use it consistently in the results.

Response: We have updated to be consistent and note aOR in each case. 

• Results: you are referring to “significantly higher “rates. Did you conduct one-tail or two tailed statistical tests? Your method section does not indicate the bivariate tests used (Line 200).

Response: We added a statement on using two-tailed in the data analysis section of the methods. We also added survey-weighted chi-square for bivariate analysis in the methods section. 

• Results: Be consistent with rounding for percentages reported though this section including tables.

Response: We have updated to remain consistent. 

• Results: In tables, improve title and include sample sizes in the title or somewhere in the table. Distinguish headers from variables. Perhaps consider moving the asterisks denoting significant findings.

Response: We have updated the tables though moving the asterisks may not reflect the significance of the cell. However, we have tabbed the variables to make it more clear. 

• Tables: Consider justifying to the right for the numeric columns.

Response: They have been updated to be justified to the right.

• Table 2: You have a symbol in front of “~U.S. Citizen” that you did not define at the footer of the table.

Response: This was a typo that has been now deleted.

• Conclusion: this section lacks detail particularly to what the results indicate. What type of clinicians are your referring to? What type of assessments are your referring to?

Response: We have updated the conclusion to clarify these items.

---

## [Editor Report · Decision Letter 2]

16 Feb 2021

Experiences of violence and mental health outcomes among Asian American transgender adults in the United States

PONE-D-20-21733R2

Dear Dr. Becerra,

We’re pleased to inform you that your manuscript has been judged scientifically suitable for publication and will be formally accepted for publication once it meets all outstanding technical requirements.

Kind regards,

Michelle Tye, Ph.D.

Academic Editor

PLOS ONE
---

## [Editor Report · Acceptance letter]

23 Feb 2021

PONE-D-20-21733R2 

Experiences of violence and mental health outcomes among Asian American transgender adults in the United States. 

Dear Dr. Becerra:

I'm pleased to inform you that your manuscript has been deemed suitable for publication in PLOS ONE. Congratulations! Your manuscript is now with our production department. 

Kind regards, 

on behalf of

Dr. Michelle Tye 

Academic Editor

PLOS ONE